# Co-Occurring Conduct Problems and Anxiety: Implications for the Functioning and Treatment of Youth with Oppositional Defiant Disorder

**DOI:** 10.3390/ijerph20043405

**Published:** 2023-02-15

**Authors:** Thorhildur Halldorsdottir, Maria G Fraire, Deborah A. G. Drabick, Thomas H. Ollendick

**Affiliations:** 1Department of Psychology, School of Social Sciences, Reykjavik University, 102 Reykjavik, Iceland; 2Centre of Public Health Sciences, Faculty of Medicine, University of Iceland, 101 Reykjavik, Iceland; 3Department of Psychiatry, McLean Hospital, Harvard Medical School, Boston, MA 02115, USA; 4Department of Psychology and Neuroscience, Temple University, Philadelphia, PA 19122, USA; 5Child Study Center, Department of Psychology, Virginia Tech, Blacksburg, VA 24060, USA

**Keywords:** oppositional defiant disorder, conduct problems, anxiety, subtypes, latent profile analysis

## Abstract

Conduct problems and anxiety symptoms commonly co-occur among youths with oppositional defiant disorder (ODD); however, how these symptoms influence functioning and treatment outcomes remains unclear. This study examined subtypes based on these co-occurring symptoms in a clinical sample of 134 youths (M_age_ = 9.67, 36.6% female, 83.6% white) with ODD and the predictive power of these subgroups for youth functioning and psychosocial treatment outcomes. The latent profile analysis (LPA) was used to identify subgroups based on parent- and self-reported conduct problems and anxiety symptoms. Differences among the subgroups in clinician-, parent-, and/or self-reported accounts of symptom severity, school performance, underlying processing known to be impaired across ODD, conduct and anxiety disorders, self-concept, and psychosocial treatment outcomes were examined. Four distinct profiles were identified: (1) Low Anxiety/Moderate Conduct Problems (*n* = 42); (2) High Anxiety/Moderate Conduct Problems (*n* = 33); (3) Moderate Anxiety/Moderate Conduct Problems (*n* = 40); and (4) Moderate Anxiety/High Conduct Problems (*n* = 19). The Moderate Anxiety/High Conduct Problems group exhibited more severe behavioral problems, greater difficulties with negative emotionality, emotional self-control, and executive functioning; they also demonstrated worse long-term treatment outcomes than the other subgroups. These findings suggest more homogeneous subgroups within and across diagnostic categories may result in a deeper understanding of ODD and could inform nosological systems and intervention efforts.

## 1. Introduction

Oppositional defiant disorder (ODD) is characterized by a developmentally inappropriate and persistent pattern of irritable mood, defiant behavior, argumentativeness, and hostile behavior towards authority figures [1]. These symptoms lead to social, emotional, and/or academic deficits during childhood, which persist into adulthood if left untreated [2]. The lifetime prevalence of ODD is estimated to be 10.2% [3]. Parent-focused treatments based on behavioral elements, such as positive reinforcement of desirable child behavior, have been identified as evidence-based treatments for ODD [4]. However, one-third to one-half of youths do not respond to these treatments [5]. Given their pervasiveness, resistance to change, and financial burden to society [6], a better understanding of the complex and heterogeneous presentation of oppositional behaviors is necessary to improve the assessment and treatment of ODD. Although the dimensional nature of ODD continues to be studied [7,8,9,10,11,12], an additional avenue to pursue is understanding treatment non-responders and the role of comorbidity and potential underlying processes that contribute to different symptom profiles. 

Comorbidity is common among youths diagnosed with ODD [13,14]. Although attention-deficit/hyperactivity disorder (ADHD) has received the most attention, conduct and anxiety disorders are also common among youths with ODD [14,15]. Indeed, clinical studies have shown that >30% of youths with ODD also meet the diagnostic criteria for CD and up to 60% meet the diagnostic criteria for an anxiety disorder [16]. These high comorbidity rates may be indicative of shared psychosocial and biological factors contributing to the disorders [17,18]. However, previous reports on how co-occurring conduct problems and anxiety influence global functioning and treatment responses to psychosocial interventions among youths with ODD have yielded mixed results [18,19,20,21,22]. These inconsistent findings suggest that co-occurring conduct problems and anxiety symptoms may be associated with different profiles of ODD with varying clinical presentations and prognosis [23]. The aim of this study was to extend this literature to a treatment-seeking sample.

The latent profile analysis (LPA) is a person-centered data analytic approach that identifies profiles across the continuum of certain behaviors. A person-centered analytic approach is particularly important in clinical research when exploring heterogeneous populations, such as children with ODD, as it is not assumed that the same processes apply to all individuals, unlike with more traditional, variable-centered analytic approaches. Furthermore, LPAs have the capacity to identify clinically meaningful subgroups from relatively small sample sizes [24,25]. Previous person-centered analyses of ODD have formed subgroups based on oppositional behaviors alone [7,8,19,26,27,28,29]. This research suggests that different dimensions of ODD may be predictive of varying comorbid difficulties later in life (e.g., [7,19]). For instance, one study found that youths with ODD characterized by primarily irritable symptoms were more likely to present with internalizing problems in late adolescence, whereas those primarily characterized by headstrong or hurtful symptoms were more likely to present with substance use [11]. Despite the high levels of comorbidity, to our knowledge, the current study is the first to examine subtypes based on these co-occurring conduct problems and anxiety symptoms in a clinical sample of youths with ODD.

The current study was designed to (1) identify subgroups based on conduct problems and anxiety symptom profiles in a clinical sample of 134 children with ODD using LPA, and (2) examine the predictive power of these subgroups to predict global functioning and treatment outcomes following empirically supported psychosocial interventions [30]. Measures of global functioning were informed by empirical research supporting their inclusion [18,31,32,33,34,35] and potential clinical implications for psychosocial interventions [36]. Specifically, we examined whether the identified subgroups differed on clinician-, parent-, and/or self-reported accounts of symptom severity and presentation (i.e., proactive and reactive aggression); school performance; underlying processing known to be impaired across ODD, CD, and anxiety disorders (i.e., lability, emotional self-control, and executive functioning [37]); self-concept; and treatment outcomes following psychosocial interventions.

## 2. Method

### 2.1. Participants

The group of participants comprised 134 7–14 year-old (Mean_age_ = 9.67, SD = 1.82) youths referred for ODD treatment (ClinicalTrials.gov NCT00510120) [30]. Informed consent and assent were obtained from families and participants, respectively. The presence of psychiatric disorders, including ODD, was determined using a semi-structured clinical interview (see below). Children who met full criteria for autism spectrum disorder, a psychotic disorder, CD, or possessed an estimated IQ < 80 were excluded from the study. Although full diagnostic criteria for CD were not met in this study, youths presented with an average of 1.93 maternally reported and 1.21 paternally reported CD symptoms on the Disruptive Behavior Disorders Rating Scale (DBDRS, see below). Participants were recruited from schools, churches, pediatricians, child psychiatrists, and medical clinics.

### 2.2. Measures

Appendix A displays the measurements completed by each informant and internal consistency of the subscales. For all the measures, higher scores reflect greater impairment, unless otherwise specified in the description below.

#### 2.2.1. Latent Profile Classification Measures

##### Parental and Teacher Ratings

The Conduct Problems (9 items; e.g., *steals, deceives others*) and Anxiety subscales (14 items; e.g., *worries, is fearful, tries too hard to please others*) of the Behavior Assessment System for Children, Second Edition (BASC-2) [38], were used to assess parent- and teacher-reported conduct problems and anxiety, respectively, in the LPA.

##### Self-Ratings

The Disruptive Behavior Disorders Rating Scale (DBDRS) [39] was used to assess self-reported symptoms of ODD and CD based on DSM-IV criteria (8 items). A floor effect was observed in self-reported CD symptoms endorsed, with only 18.6% (*n* = 25) endorsing >1 symptom. Therefore the ODD subscale, which had greater variability in responses, was used in the LPA. The Anxiety subscales of the Beck Youth Inventories, Second Edition (BYI-2) [40], was used to measure self-reported anxiety (20 items; e.g., *I worry*, *I am afraid something bad might happen to me*).

#### 2.2.2. External Validator Measures

##### Clinician Ratings

Diagnostic interview. The Anxiety Disorders Interview Schedule for DSM-IV, Child and Parent Versions (ADIS-C/P) [41], are semi-structured diagnostic interviews in which clinicians assign a clinical severity rating (CSR) on a 9-point scale (0–8, with a rating ≥ 4 suggesting a clinical level of interference). See Appendix A for further description of the administration and psychometric properties of the ADIS C/P. The CSRs derived from the ADIS served as a clinician-rated measure of severity of anxiety disorder, attention-deficit/hyperactivity disorder (ADHD), and ODD prior to and following treatment.

Global functioning. The Child Global Assessment Scale (CGAS) [42] is a 100-point rating scale measuring psychological, social, and school functioning in youths, with lower scores reflecting greater impairment. Participants’ global functioning was assessed prior to and following treatment.

##### Parental Ratings

ODD and CD symptoms. The ODD and CD subscales of the parent version of the Disruptive Behavior Disorders Rating Scale (DBDRS) [39] were used to measure parent-reported ODD and CD severity. Reflecting DSM-IV symptoms, the ODD scale contains 8 items rated on a four-point Likert scale (0 = *never* to 3 = *very often*), whereas the CD scale consists of 15 dichotomous questions in which an item is either endorsed as present or absent by the parent.

Aggression and emotional self-control. Parental ratings on the Aggression and Emotional Self-Control subscales from the BASC-2 [38] were used. The Aggression subscale (11 items; e.g., *bullies others, threatens to hurt others*) captures general aggressive behaviors towards peers and parents. The Emotional Self-Control subscale assesses difficulties regulating emotions and affect (6 items; e.g., *acts out of control, has poor self-control*).

Lability. The Lability/Negativity scale (11 items; e.g., *exhibits wide mood swings, responds negatively to neutral or friendly overtures by peers*) of the Emotion Regulation Checklist (ERC) [43] was used to measure dysregulated negative affect, lability, and inflexibility.

Proactive and reactive aggression. The Proactive Aggression (10 items; e.g., has hurt others to win a game or contest, gets others to gang up on children) and Reactive Aggression subscales (6 items; e.g., gets mad when corrected, blames others when gets into trouble) of the Child Behavior Rating Scale (CBRS) [44] were utilized to assess parental reported proactive (i.e., aggressive behavior intended to harm or coerce another person) and reactive (i.e., defensive response to perceived threat or provocation) aggression.

Executive functioning. The Global Executive Composite (86 items) of the Behavior Rating Inventory of Executive Functioning (BRIEF) [45] was used to assess executive functioning in daily living.

##### Teacher Ratings

Aggression, school problems, social skills, learning problems, and study skills. Teacher ratings on the Aggression, School Problems, Learning Problems, and Study Skills subscales of the BASC-2 [38] were used to examine aggression, academic problems, and school functioning. Higher scores on the Aggression, School Problems, and Learning Problems subscales indicate greater impairment, whereas lower scores on the Study Skills subscale reflect impaired abilities.

##### Self-Ratings

Proactive and reactive aggression. A parallel version of the CBRS [44] was administered to assess self-reported proactive and reactive aggression.

Self-concept. The Self-Concept subscale of the BYI-2 [40] was used to assess self-perceived competence and positive self-worth (20 items; e.g., *I feel smart, I like myself*). Higher scores reflect a more positive self-concept.

## 3. Treatment

Participants received 12 sessions of Parent Management Training (PMT) [46] or Collaborative & Proactive Solutions (CPS) [47]. The PMT protocol was based on Barkley’s manualized training program [46]. The treatment consists of providing parents with psychoeducation on the causes of defiant, non-compliant behavior and teaching parents how to (1) implement positive attending through the use of “special time”; (2) use attending skills and effective commands to increase compliant behavior; (3) implement a contingency management program; (4) use the time-out procedure. The CPS protocol was based on Greene’s CPS model [47]. The CPS program builds on teaching parents how to solve problems collaboratively and proactively with their child. This involves teaching parents to (1) identify lagging skills and unsolved problems that contribute to defiant, non-compliant behaviors; and (2) techniques to solve these problems with their child. Both treatments have been found effective in treating ODD and have yielded equivalent and positive treatment outcomes [30,48]. Details of the treatment protocols and procedure are described elsewhere [30].

## 4. Statistical Analysis

The LPA is based on a step-wise procedure to identify profiles [49]. The Akaike Information Criterion (AIC) [50], Bayesian Information Criterion (BIC) [51], sample size adjusted BIC (ABIC) [52], Bootstrap Likelihood Ratio Test (BLRT), and entropy were used to determine the best fitting model. See Appendix A for further details on the LPA, model fit indices, and descriptive statistics of the conduct problems and anxiety symptoms variables included in the LPA (Appendix A).

After selecting the best-fitting model, tests of equality of means across latent profiles were conducted to examine whether profiles differed on demographic variables. In this procedure, class membership is held constant and a chi-square statistic for omnibus and pairwise comparisons across latent profiles is provided. If the omnibus tests are significant, the pairwise comparisons are explored. Given the non-normal distribution of several teacher-reported external validators (i.e., learning problems, study skills, and general aggression), non-parametric statistics were used to examine group differences. Raw scores were used for all analyses and α < 0.05 was considered significant. The Full Information Maximum Likelihood estimation was used to address missing data (see Appendix A). Analyses were conducted using Mplus Version 7.1 [49].

## 5. Results

Table 1 displays demographics and clinical characteristics of the sample.

### 5.1. Latent Profile Classification

LPA was conducted to derive discrete profiles based on conduct problems and anxiety symptoms. Table 2 illustrates the fit indices based on the different models.

Although the BIC increased with the three- and four-class models, the sample-size adjusted BIC continued to decrease, as did the AIC, and the BLRT *p*-value was significant, indicating that the model fit improved with the addition of each class. A five-class model failed to converge owing to local maxima (likely outliers that were unduly influencing class membership) and thus is not presented. As such, a four-class solution best fits the data (BIC = 5427.67, AIC = 5303.06, ABIC = 5291.65; see Figure 1 and Appendix A).

In the four-class solution, Class 1 was characterized by lower levels of anxiety reported across informants relative to the other classes; however, the level of conduct problems was comparable to the other classes (Low Anxiety and Moderate Conduct Problems Class, Low Anx/Mod CP; *n* = 42). Class 2 was distinctive due to elevated levels of parent-reported anxiety symptoms and moderate conduct problems across informants (High Anxiety and Moderate Conduct Problems Class, High Anx/Mod CP; *n* = 33). Individuals in Class 3 obtained moderate levels of both conduct problems and anxiety symptoms across informants (Moderate Anxiety and Moderate Conduct Problems Class, Mod Anx/Mod CP; *n* = 40). Class 4 was characterized by high levels of conduct problems and moderate anxiety symptoms (Moderate Anxiety and High Conduct Problems Class, Mod Anx/High CP; *n* = 19).

### 5.2. Global Functioning and Treatment Outcomes

Table 3 displays means, standard deviations, and profile differences for external validators.

#### 5.2.1. Clinician Ratings

Prior to treatment, no profile differences in the ODD, anxiety disorder, and ADHD CSRs were noted. However, the two profiles with high symptom severity (High Anx/Mod CP and Mod Anx/High CP) were rated as more impaired overall, as measured using the CGAS, than the other two profiles.

The profiles did not differ in terms of the proportion of youths receiving each kind of treatment (*p* = 0.73). Comparable treatment outcomes were noted at post-treatment and the 6-month follow-ups across the profiles. However, at the 1-year follow-up, children in the Mod Anx/High CP class had significantly higher levels of ODD symptoms and lower clinician-rated global functioning than the children in the other profiles.

#### 5.2.2. Parental Ratings

The profiles differed on maternal and paternal ratings of lability and emotional self-control, with higher levels of negative emotionality and difficulties regulating emotions and affect among youths in the High Anx/Mod CP and Mod Anx/High CP profiles compared to the Low Anx/Mod CP profile. Children in the Mod Anx/Mod CP profile were also rated by mothers as having greater difficulties with emotional self-control than children in the Low Anx/Mod CP profile.

Youths in the High Anx/Mod CP and Mod Anx/High CP profiles displayed higher levels of ODD and CD symptoms and aggression and displayed greater executive functioning deficits based on paternal reports than youths in the other profiles. Furthermore, higher levels of paternal-reported proactive aggression were noted among youths within the Mod Anx/High CP profile than those in the other profiles.

#### 5.2.3. Teacher Ratings

Teachers rated the youths in the Mod Anx/High CP profile as the most impaired overall, with the highest levels of aggression, school problems, and learning problems, as well as the lowest study skills. Children in the Low Anx/Mod CP profile also had lower social and study skills than youths in the High Anx/Mod CP and Mod Anx/Mod CP profiles.

#### 5.2.4. Self-Ratings

Youths within the Mod Anx/High CP profile reported higher levels of proactive and reactive aggression and lower self-concept than children in the other profiles.

## 6. Discussion

To the best of our knowledge, this study is the first to examine the differential clinical presentation and treatment outcomes of youths with ODD based on co-occurring conduct problems and anxiety symptoms using a person-centered approach. The LPA revealed four subgroups that differed across various areas of functioning, as well as long-term treatment outcomes.

Overall, children with high levels of conduct problems and moderate levels of anxiety symptoms (i.e., the Mod Anx/High CP subgroup) were the most impaired. Youths within this profile were the only ones that exhibited severe behavioral problems across multiple informants. They were also rated as having greater difficulties associated with negative emotionality, emotional self-control, and executive functioning by at least one informant. Although an initial symptom reduction was observed, children within this profile returned to similar levels of pre-treatment ODD symptoms at the 1-year follow-up, whereas treatment gains were largely maintained in the other profiles. This finding is consistent with other recent person-centered analyses on the poor prognosis of youths with high levels of conduct problems [7,8]. These collective findings suggest that augmented treatment may be necessary for youths in this subgroup if treatment gains are to be sustained over time. This approach may include parallel treatment for the anxiety symptoms or more transdiagnostic approaches that target broader underlying processes such as emotion regulation or executive functioning skills. Future studies should investigate whether adding such components yield better long-term outcomes.

A second profile emerged with comparable levels of anxiety as this impaired profile (i.e., Mod Anx/Mod CP subgroup). However, despite similar levels of anxiety, youths within the Mod Anx/Mod CP subgroup appeared to be higher functioning across multiple areas compared to the Mod Anx/High CP subgroup. Indeed, children within this subgroup displayed lower levels of behavioral problems across informants and were rated as having higher levels of effortful control and better social and study skills compared to children in the Mod Anx/High CP subgroup. Youths within the Mod Anx/Mod CP subgroup showed significant treatment gains that were maintained over time. This finding suggests that targeting behavioral problems for children with this type of symptom presentation is an effective treatment option.

The High Anx/Mod CP profile differed from the other profiles as they exhibited greater levels of anxiety symptoms according to their parents. Although slightly less impaired, youths within this subgroup were similar to the children in the Mod Anx/High CP subgroup. Specifically, children within this profile presented with elevated levels of conduct problems according to their fathers and also reportedly exhibited difficulties with emotion regulation and executive functioning based on parental reports. In terms of strengths, youths in the High Anx/Mod CP subgroup were rated as having higher study skills than children within the other profiles. Additionaly, unlike the Mod Anx/High CP subgroup, children in the High Anx/Mod CP subgroup maintained treatment gains at the 1-year follow-up. Theis finding suggests that current psychosocial treatments are generally effective for youths with this symptom presentation despite some functional deficits, similar to those observed in the multiple problem subgroup.

One profile was defined by low anxiety symptoms (the Low Anx/Mod CP class). Interestingly, youths within this profile were generally comparable in terms of overall functioning and treatment outcomes to the children in the Mod Anx/Mod CP subgroup, with a few exceptions. For instance, youths within this group had lower social and study skills according to their teachers than the Mod Anx/Mod CP profile.

Notably, several theories have been put forth to explain the mixed presentation of comorbid behavioral problems and anxiety among youths [23,53,54,55]. When comparing these theories, the two classes with comparable anxiety levels (i.e., Mod Anx/Mod CP and Mod Anx/High CP) in this study are perhaps most consistent with the dual-pathway model [23]. Specifically, Drabick and colleagues [23] proposed that anxiety mitigates the expression of ODD (i.e., the buffering hypothesis) among youths with low anger/frustration, moderate-to-high fear, high levels of proactive aggression, and age-appropriate levels of effortful control, executive functioning, and social information processing. Alternatively, they suggested anxiety exacerbates (i.e., the multiple-problem hypothesis) the expression of ODD among youths with high anger/frustration, low fear, low effortful control, high levels of reactive aggression, and poor executive functioning and interpersonal competence.

Another interesting finding in this study pertained to informant discrepancies in behavioral ratings. In accordance with previous studies (e.g., [56]), mothers generally provided higher ratings of problematic behaviors across subgroups than other informants. As a result, less variation between groups on external validators was observed based on maternal ratings than other informants. Such discrepancies in ratings have been attributed to the context in which the child is observed, child characteristics, and the rater’s attribution of the child’s behavior [57]. Thus, the addition of other informants yielded important clinical information. For instance, youths in the Mod Anx/High CP subgroup, who exhibited severe behavioral problems in the school setting according to teacher reports, had the worst treatment outcomes. This finding highlights the need to use a multi-informant approach when assessing behavioral problems to obtain a holistic picture of the child’s functioning and to inform treatment options.

Although the study possesses several strengths, such as using a multi-informant person-centered approach and the longitudinal assessment of treatment outcomes, there are several weaknesses. First, youths meeting the diagnostic criteria for CD were excluded from this study. However, many of the youths presented with subthreshold levels of CD symptoms (see Table 1), providing sufficient variability to conduct the LPA. Secondly, subgroups were based on conduct problems and anxiety symptoms, which are understudied but common among youths with ODD. Person-centered studies on how other commonly co-occurring psychiatric symptoms, such as ADHD symptoms, influence functioning and treatment outcomes of youths with ODD remain an important venture for future research. Finally, future studies are also needed to examine the longer-term trajectories of these subtypes. This study did provide some indication of prognosis by tracking participants for one year after treatment completion. However, further follow-up could be informative for prevention and treatment development to examine, for example, whether certain subtypes are more likely to develop depression and/or CD than others.

## 7. Conclusions

In conclusion, research on how conduct problems and anxiety symptoms influence functioning and treatment outcomes of youths with ODD has been scarce. Using a person-centered approach, we identified subtypes of ODD based on co-occurring conduct problems and anxiety that meaningfully differed in clinical presentation and treatment prognosis. This research highlights the need for determining more homogeneous subgroups within and across diagnostic categories to gain a deeper understanding of ODD, inform future nosological systems, and enhance intervention efforts.

## Figures and Tables

**Figure 1 ijerph-20-03405-f001:**
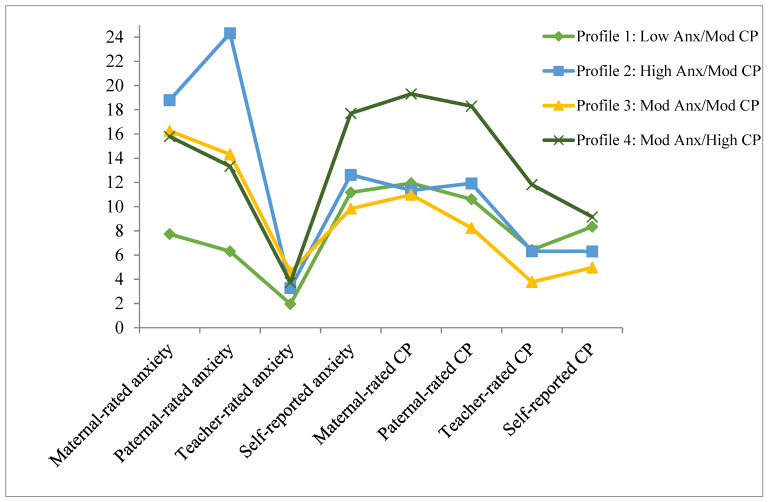
Average anxiety and conduct problems (CP) for each class in the four-profile model. Note. Anx = anxiety, CP = conduct problems, mod = moderate.

**Table 1 ijerph-20-03405-t001:** Demographics and clinical characteristics of the whole sample.

Categorical Variables	*N (%)*
Gender	
Female	49 (36.6)
Male	85 (63.4)
Race	
White	112 (83.6)
Non-white	22 (16.4)
Comorbid anxiety disorder	
Generalized anxiety disorder	30 (22.39)
Separation anxiety disorder	16 (11.94)
Social phobia	23 (17.16)
Specific phobia	26 (19.40)
Comorbid ADHD	89 (66.4)
Continuous Variables	M (SD)
Age in years	9.67 (1.82)
ODD CSR	5.99 (1.05)
CGAS	59.37 (5.89)
Maternal-reported CD symptoms	1.93 (1.74)
Paternal-reported CD symptoms	1.21 (1.44)

Note. *N* = sample size, CSR = clinician severity rating, CGAS = children’s global assessment scale.

**Table 2 ijerph-20-03405-t002:** Fit indices for LPA Models with 1 to 4 classes.

Number of Classes	Number of Free Parameters	Log Likelihood	AIC	BIC	ABIC	BLRT
1	16	−2674.35	5380.70	5427.07	5376.46	-
2	25	−2644.15	5338.30	5410.74	5331.66	<0.001
3	34	−2623.92	5315.84	5414.37	5306.82	<0.001
4	43	−2608.53	5303.06	5427.67	5291.65	0.0128

Note. ABIC = Adjusted Bayesian Information Criterion; AIC = Akaike Information Criterion; BIC = Bayesian Information Criterion; BLRT = Bootstrap Likelihood Ratio Test (*p*-value).

**Table 3 ijerph-20-03405-t003:** Comparison of external validators in the four-profile model.

	Profile 1 Low Anx/Mod CP		Profile 2 High Anx/Mod CP		Profile 3 Mod Anx/Mod CP		Profile 4 Mod Anx/High CP		Pairwise
	M	SD	M	SD	M	SD	M	SD	Comparisons
Clinician ratings									
ANX CSR pre	2.82	2.68	4.05	2.51	3.22	2.67	4.35	2.43	-
ODD CSR pre	5.81	1.12	6.18	1.01	5.86	1.20	6.30	1.18	-
ODD CSR post	3.85	2.83	3.52	2.76	3.18	2.73	4.37	3.09	-
ODD CSR 6-month	3.89	3.74	3.46	3.48	3.14	2.59	4.95	2.53	-
ODD CSR 1-year	3.71	3.54	3.94	4.19	2.98	3.00	6.16	2.38	4 > 1, 2, 3
CGAS pre	60.10	5.64	57.80	6.31	61.50	7.14	56.60	5.01	1, 3 > 4; 3 > 2
CGAS post	67.18	10.41	67.41	8.77	68.99	11.63	63.89	8.96	-
CGAS 6-month	66.44	16.82	67.73	17.61	68.66	12.45	51.56	20.81	-
CGAS 1-year	66.66	11.96	65.29	17.73	72.00	15.25	57.22	15.15	1, 3 > 4
Maternal Ratings									
ODD symptoms	5.42	2.05	5.74	2.03	5.63	2.01	6.43	2.31	-
CD symptoms	1.87	1.92	1.84	1.78	1.68	1.95	2.57	2.55	-
Externalizing problems	71.40	11.60	72.30	9.30	69.00	10.90	85.30	10.30	4 >1, 2, 3
Aggression	69.79	12.76	71.01	10.99	68.47	11.62	77.56	16.15	-
Lability	35.10	6.87	38.60	6.14	36.60	6.07	40.2	6.98	2, 4 > 1
Negative emotionality	10.25	2.28	11.64	1.98	11.17	2.19	11.78	3.02	-
Emotional self-control	8.45	3.56	10.90	3.56	9.94	3.54	12.30	3.27	2, 4 > 1; 4 > 3
Proactive aggression	16.23	3.86	16.60	3.27	16.45	4.07	19.06	3.14	-
Reactive aggression	14.45	2.26	15.55	2.09	15.26	2.02	15.79	2.09	-
Executive functioning	67.55	10.63	71.36	9.74	69.25	8.96	73.54	11.73	-
Self-regulation	41.62	7.03	43.52	6.18	42.14	7.29	43.16	10.27	-
Paternal Ratings									
ODD symptoms	3.89	2.59	5.35	2.87	3.56	2.46	5.71	3.88	2 > 1, 3; 4 > 3
CD symptoms	1.04	1.54	1.78	2.42	.72	1.35	2.41	2.45	2 > 3; 4 > 3, 1
Externalizing problems	65.00	8.88	71.90	11.00	62.00	10.40	81.70	13.30	4 > 2 > 1, 3
Aggression	63.10	9.72	68.60	12.30	61.40	9.92	73.50	17.90	2, 4 > 1, 3
Lability	33.90	7.13	38.90	6.03	35.00	7.52	40.80	11.30	2, 4 > 1, 3
Negative emotionality	9.00	2.53	11.00	2.07	8.99	2.59	10.10	4.53	2 > 1, 3
Emotional self-control	7.63	3.24	10.60	3.44	8.17	3.16	10.20	4.84	2 > 1, 3; 4 > 1
Proactive aggression	15.00	3.18	15.80	3.16	15.00	4.55	19.30	4.71	4 > 1, 2, 3
Reactive aggression	13.59	2.85	14.91	2.81	14.02	3.18	16.02	3.46	-
Executive functioning	64.20	12.20	70.30	10.10	65.30	11.90	72.60	15.60	2, 4 > 1
Self-regulation	36.30	9.78	41.50	8.95	37.10	9.23	42.90	14.50	2 > 1, 3
Teacher Ratings									
Externalizing problems	59.90	16.30	59.10	18.00	53.50	15.60	72.00	17.80	4 > 1, 2, 3
Aggression	61.00	19.50	59.10	22.70	53.90	16.40	67.80	22.00	4 > 3
School problems	53.10	9.72	54.70	13.50	51.60	13.80	63.60	9.64	4 > 1, 2, 3
Social skills	38.20	13.40	45.40	14.20	47.20	16.00	40.50	13.70	2, 3 > 1
Learning problems	48.60	10.60	52.30	15.30	49.20	12.30	60.00	13.50	4 > 1, 3
Study skills	42.20	11.20	45.40	14.20	48.00	12.70	36.80	10.60	3 > 1, 4; 2 > 4
Self-Report									
Proactive aggression	12.60	2.72	12.10	2.53	11.60	2.21	14.7	3.49	4 > 1, 2, 3
Reactive aggression	8.85	2.27	8.92	2.41	8.29	2.40	10.7	3.05	4 > 1, 2, 3
Self-concept	40.10	11.80	42.90	10.70	44.60	11.30	34.9	13.70	2, 3 > 4

Note. ANX = anxiety disorder; CD = conduct disorder; CGAS = Child Global Assessment Scale; CP = conduct problems; CSR = clinician severity rating; Mod = moderate; ODD = oppositional defiant disorder.

## Data Availability

The data in this study are not publicly available due to patient confidentiality.

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
