# Peer review of "Co-Occurring Conduct Problems and Anxiety: Implications for the Functioning and Treatment of Youth with Oppositional Defiant Disorder"

_ijerph, 2023, doi:10.3390/ijerph20043405_

Round 1

Reviewer 1 Report

The work presented shows clarity in the contents and I think it is a very interesting investigation. However, I suggest a few suggestions as an improvement to the document: At lines 51-52, the authors say: "Recent research on the dimensions of ODD and comorbidity also suggests that different dimensions may be predictive of different psychopathologies (eg, 5,17)." I think I could give an example to Expand information.

It would be advisable to add some more up-to-date quotes into the discussion that reinforce the results obtained.

Author Response

Reviewer 1:

The work presented shows clarity in the contents and I think it is a very interesting investigation. However, I suggest a few suggestions as an improvement to the document: At lines 51-52, the authors say: "Recent research on the dimensions of ODD and comorbidity also suggests that different dimensions may be predictive of different psychopathologies (eg, 5,17)." I think I could give an example to Expand information.

Thank you for your comments. Here is a point-by-point response to both comments in italics. In response to this comment, we have provided an example of findings as illustrated here below. Note however that we have moved the text to lines 74-80 in order to maintain flow within the text.

“Previous person-centered analyses of ODD have formed subgroups based on oppositional behaviors alone [7,8,19,26–29]. This research suggests that different dimensions of ODD may be predictive of varying comorbid difficulties later in life (e.g., [7,19]). For instance, one study found that youth with ODD characterized by primarily irritable symptoms were more likely to present with internalizing problems in late adolescence, whereas those primarily characterized by headstrong or hurtful symptoms were more likely to present with substance use [11].”

It would be advisable to add some more up-to-date quotes into the discussion that reinforce the results obtained.

We appreciate this comment and we have strived to quote the most recent relevant findings. Specifically, prior to submitting the paper, we conducted a literature review of studies using a person-centered approach to examine subtypes of ODD. Thus, we believe that we do cite the most recent relevant literature in this field in the discussion.

Reviewer 2 Report

This study deals with a topical and relevant subject, using also a novel and not very widespread methodological perspective, which gives the article a considerable scientific interest.

Introduction

The introduction is descriptive of the problem and specific, although perhaps some aspects could be completed. For example, evidence-based treatments for ODD are mentioned, although it is mentioned that between a third and a half of patients do not respond to it. Perhaps it would be interesting to briefly describe what this treatment consists of (do we understand that it refers to cognitive-behavioral therapy?). On the other hand, perhaps it would be good to also mention the data on the prevalence of the disorder in the introduction.

Method
The explanations of the method, sample, and different instruments used are adequate and allow replication of the study. However, Table 1, which appears in Section 5 of the results, could be better inserted after the paragraph on description in the section on participants.

Treatment
The section on treatment is inadequate, as it is at least necessary to briefly summarise what these treatments consist of. Also, the claim that both types of treatments have equivalent and positive results needs a bibliographic reference to back it up.

Results
The results and discussion are coherent and correctly presented.

Bibliographic references

The citation system is incorrect because the citations are not in square brackets, as the journal guidelines require, but in "footnotes." There is numerical duplication in the works cited, so the citation number appears twice.

Author Response

Reviewer 2:

This study deals with a topical and relevant subject, using also a novel and not very widespread methodological perspective, which gives the article a considerable scientific interest.

Thank you for your comments. Here below is a point-by-point response to each comment in italics.

Introduction

The introduction is descriptive of the problem and specific, although perhaps some aspects could be completed. For example, evidence-based treatments for ODD are mentioned, although it is mentioned that between a third and a half of patients do not respond to it. Perhaps it would be interesting to briefly describe what this treatment consists of (do we understand that it refers to cognitive-behavioral therapy?). On the other hand, perhaps it would be good to also mention the data on the prevalence of the disorder in the introduction.

We have added the following sentence to the introduction regarding evidence treatments for ODD (line 37-39) and the lifetime prevalence of ODD (line 37-39):

Parent-focused treatments based on behavioral elements, such as positive reinforcement of desirable child behavior, have been identified as evidence-based treatments for ODD [4].”

The lifetime prevalence of ODD is estimated to be 10.2% [3]. (line 37)

Method
The explanations of the method, sample, and different instruments used are adequate and allow replication of the study. However, Table 1, which appears in Section 5 of the results, could be better inserted after the paragraph on description in the section on participants.

 We agree that Table 1 could also nicely fit after the description of participants in the Methods section. We will however leave the decision on where to place the Table in the hands of the IJERPH’s processing team in order to be consistent with the journal’s format.

Treatment
The section on treatment is inadequate, as it is at least necessary to briefly summarise what these treatments consist of. Also, the claim that both types of treatments have equivalent and positive results needs a bibliographic reference to back it up.

In response to this comment, we have added the brief summary below of the treatment to the introduction (lines 189-199). We also reference two independent studies (Murrihy et al., 2022 and Ollendick et al., 2016) illustrating a comparable decrease in ODD symptoms among youth after Parent Management Training and Collaborative & Proactive Solutions.

“Participants received 12 sessions of Parent Management Training (PMT)[46] or Collaborative & Proactive Solutions (CPS) [47]. The PMT protocol was based on Barkley’s manualized training program [46]. The treatment consists of providing parents with psychoeducation on the causes of defiant, non-compliant behavior and teaching parents how to 1) implement positive attending through the use of “special time”; 2) use attending skills and effective commands to increase compliant behavior; 3) implement a contingency management program; 5) use the time-out procedure. The CPS protocol was based on Greene’s CPS model [47]. The CPS program builds on teaching parents how to solve problems collaboratively and proactively with their child. This involves teaching parents to 1) identify lagging skills and unsolved problems that contribute to defiant, non-compliant behaviors; and 2) techniques to solve these problems with their child.”

Results
The results and discussion are coherent and correctly presented.

Bibliographic references

The citation system is incorrect because the citations are not in square brackets, as the journal guidelines require, but in "footnotes." There is numerical duplication in the works cited, so the citation number appears twice.

We have gone over the citations and added square brackets to be consistent with the journal’s guidelines and the duplicate numbers have been removed.